# Harnessing the Bioactive Potential of *Limonium spathulatum* (Desf.) Kuntze: Insights into Enzyme Inhibition and Phytochemical Profile

**DOI:** 10.3390/plants12193391

**Published:** 2023-09-26

**Authors:** Seria Youssef, Luisa Custódio, Maria João Rodrigues, Catarina G. Pereira, Ricardo C. Calhelha, József Jekő, Zoltán Cziáky, Karim Ben Hamed

**Affiliations:** 1Laboratory of Extremophile Plants, Center of Biotechnology of BorjCedria, Hammam-Lif 2050, Tunisia; seriayoussef93@gmail.com; 2Centre of Marine Sciences (CCMAR), Universidade do Algarve, Campus de Gambelas, 8005-139 Faro, Portugal; lcustodio@ualg.pt (L.C.); mjrodrigues@ualg.pt (M.J.R.); cagpereira@ualg.pt (C.G.P.); 3Centro de Investigação de Montanha (CIMO), Instituto Politécnico de Bragança, Campus de Santa Apolónia, 5300-253 Bragança, Portugal; calhelha@ipb.pt; 4Agricultural and Molecular Research and Service Institute, University of Nyíregyháza, 4405 Nyíregyháza, Hungary; jeko.jozsef@nye.hu (J.J.); cziaky.zoltan@nye.hu (Z.C.)

**Keywords:** sea lavender, enzyme inhibitors, phenolic compounds, cytotoxicity

## Abstract

This study assessed the halophyte species *Limonium spathulatum* (Desf.) as a possible source of natural ingredients with the capacity to inhibit enzymes related to relevant human health disorders and food browning. Extracts using food-grade solvents such as water and ethanol were prepared by maceration from dried *L. spathulatum* leaves. They were evaluated for in vitro inhibition activity of enzymes such as acetylcholinesterase (AChE) and butyrylcholinesterase (BChE), α-glucosidase, tyrosinase and lipase, related to Alzheimer’s disease, type-2-diabetes mellitus, skin hyperpigmentation, and obesity, respectively. These extracts were also appraised for in vitro acute toxicity on tumoral and non-tumoral cell lines and their chemical composition by high-performance liquid chromatography coupled with electrospray ionization mass spectrometry (HPLC-ESI-MS/MS). The extracts were more effective towards BChE than AChE. The best results were obtained with the hydroethanolic and water extracts, with IC_50_ values of 0.03 mg/mL and 0.06 mg/mL, respectively. The hydroethanolic extract had the highest capacity to inhibit α-glucosidase (IC_50_: 0.04 mg/mL), higher than the positive control used (acarbose, IC_50_ = 3.14 mg/mL). The ethanol extract displayed the best inhibitory activity against tyrosinase (IC_50_ = 0.34 mg/mL). The tested samples did not inhibit lipase and exhibited low to moderate cytotoxic activity against the tested cell lines. The hydroethanolic extract had a higher diversity of compounds, followed by the ethanol and water samples. Similar molecules were identified in all the extracts and were mainly hydroxybenzoic acids, hydroxycinnamic acids, and flavonoids. Taken together, these results suggest that *L. spathulatum* should be further explored as a source of bioactive ingredients for the food, cosmetic, and pharmaceutical industries.

## 1. Introduction

Nowadays, individuals from various regions around the globe continue to incorporate wild plants into their diets and wellness routines to enhance their health [1]. Indeed, the utilization of medicinal wild plants is increasingly gaining popularity as a growing trend in contemporary societies that are in search of nourishing food and herbal products, specifically those with properties that promote health enhancement [2]. In addition, researchers emphasize exploring natural sources to discover novel products that have the potential to serve as beneficial commodities for health promotion or as valuable leads for innovative drugs. For instance, several natural-based enzyme inhibitors, such as orlistat (pancreatic lipase inhibitor), acarbose (α-glucosidase), and galantamine (AChE inhibitor), have already been used to treat some major diseases, such as type-2-diabetes mellitus (T2DM), Alzheimer’s disease (AD), and obesity [3,4,5,6,7]. This is primarily driven by their considerable structural diversity and the general perception that they tend to be safer compared with chemically synthesized compounds. Therefore, screening of potential enzyme inhibitors from further natural sources, such as extremophile halophyte plants, is of paramount importance [8].

Halophytes are plants growing spontaneously in saline soils, containing salt concentrations of up to 200 mM of sodium chloride (NaCl) [9,10]. They withstand diverse abiotic constraints in addition to salinity, such as drought, extreme temperatures, and light intensity by employing adaptive mechanisms, including potent antioxidant defense systems consisting of enzymes (e.g., superoxide dismutase, glutathione peroxidase) [11]) and secondary metabolites (e.g., phenolics and flavonoids) [12,13,14,15]). These secondary metabolites exhibit significant biological activities crucial for improving human health, such as enzymatic inhibition.

Tunisian marine areas are home to several halophytic species, such as those belonging to the *Limonium* genus, containing high levels of bioactive secondary metabolites, including phenolic acids and flavonoids. *Limonium* species (sea lavenders) are widely distributed in the Mediterranean region [16,17,18,19]. In Tunisia, 26 endemic species are found, including *Limonium spathulatum* (Desf.) Kuntze [20]. Recent studies on this species demonstrated that *L. spathulatum* leaves are a good source of minerals and fibers, contain high levels of secondary metabolites, such as phenolics and flavonoids, and display relevant antioxidant and enzyme inhibitory properties [21,22,23]. The current study aims to expand upon existing information by evaluating the ability of the food-grade extracts from the leaves of *L. spathulatum* to inhibit some enzymes known for their regulation of some major metabolic and neurogenerative diseases. The target enzymes are acetylcholinesterase (AChE), butyrylcholinesterase (BuChE), α-glucosidase, tyrosinase, and lipase. These extracts were also evaluated for acute toxicity on a panel of tumor and non-tumor mammalian cell lines. Finally, the chemical profile of the extracts was established by high-performance liquid chromatography coupled with electrospray ionization mass spectrometry (HPLC-ESI-MS/MS). Using food-grade extracts instead of toxic solvents is crucial for extracting bioactive compounds from plants and utilizing them as a source of safe and beneficial products. This approach promotes sustainability and responsible utilization of plant-derived compounds, fostering a healthier and eco-friendly product development process.

## 2. Results and Discussion

### 2.1. Enzymatic Inhibition

The enzyme inhibitory effect of *L. spathulatum* extracts was evaluated towards AChE, BChE, tyrosinase, α-glucosidase, and lipase, and results are depicted in Table 1.

The neurotransmitter acetylcholine (ACh) is an essential chemical messenger involved in signal transduction linked to memory and learning capacities [24,25]. AChE (EC 3.1.1.7) and BChE (EC 3.1.1.8) enzymes hydrolyze ACh, restoring cholinergic neurons. BuChE is a non-selective cholinesterase that can degrade ACh and butyrylcholine, making it a substitute for AChE, which specifically catalyzes ACh. Normal cholinergic signal transduction related to learning and memory depends on ACh [26,27], and one of the hallmarks of AD is ACh deficiency [24]. The use of cholinesterase inhibitors (AChEIs) can increase the concentration of ACh and are currently the only clinical drugs approved for the therapy of AD patients [28]. In this study, the water and hydroethanolic extracts inhibited the enzyme BChE more than the enzyme AChE (Table 1). Working with methanol and methanol/water extracts from leaves of the same species collected in Algeria, Mazouz et al. [22] observed the opposite inhibitory effect. Infusions and decoctions from flowers of a related species from southern Portugal, *L. algarvense*, were also more active towards AChE [29]. Such differences may be ascribed to genotypic and climatic differences. AChE plays a main function in regulating ACh levels; however, the predominance of BChE increases in patients with AD, especially at the latter stages of the disease, where the brain has a deficit of ACh [24,30,31]. Therefore, BChE is considered a sound therapeutic target for increasing ACh levels in AD patients [24]. Several AChEIs were already isolated from natural sources, resulting in the development of therapeutic agents for the treatment of AD, such as acarbose [32]. However, the number of BChE inhibitory therapeutics from natural sources is significantly lower [33]. Our results indicate that *L. spathulatum* may be a source of selective BChE inhibitors, which are of high interest for the treatment of more advanced AD cases [30,31]. The observed cholinesterase inhibition may be related to the polyphenolic content of the extracts since previous studies established a positive correlation between the inhibition of AChE and the total phenolic content of the extracts [34,35]. Specifically, the is evidence that flavonoids, such as myricetin and quercetin, have relevant anticholinesterase inhibitory properties [36].

Tyrosinase (EC. 1.14.18.1) is a multifunctional copper-containing enzyme [37,38] vital for melanogenesis and enzymatic browning. This enzyme also participates in several biological processes in arthropods, such as wound healing, cuticle sclerotization, and protective encapsulation [39]. Tyrosinase inhibitors are therefore relevant for medical and cosmetic purposes, managing conditions related to excessive melanin production, in agricultural industries as pest control agents, and in the food industry as anti-browning ingredients [40,41]. Over the past decade, some natural tyrosinase inhibitors such as kojic acid and arbutin were identified, and the only cosmetic ingredient that can be obtained synthetically is hydroquinone [42,43]. However, some of these drugs exhibited low efficacy and toxic effects, thus urging the need to identify new molecules/products with reduced side effects and higher efficacy, which may include polyphenol-rich extracts [44,45]. In this study, all types of extracts showed an inhibitory effect on tyrosinase, although lower than arbutin (Table 1). Relevant tyrosinase inhibition was also observed in methanol and aqueous leaf extracts of *L. delicatulum* and *L. quesadense* [46] and in water extracts from leaves of *L. algarvense* [29]. Some synthetic tyrosinase inhibitors are based on natural flavonoid skeletons, providing an effective scaffold for the development of novel molecules [35]. For example, flavonoids containing a keto group (e.g., kaempferol and quercetin) displayed potent tyrosinase inhibition due to their capacity to chelate copper in the enzyme active site [47]. In this sense, the highest activity observed in the ethanol extract of *L. spathulatum* may be ascribed to the presence of flavonoids containing keto groups, such as dihydro kaempferol, naringenin, and derivatives. Additionally, other studies indicated that the number and location of the hydroxyl group on the flavonoids affected tyrosinase inhibition. For example, the number of hydroxyl groups on the B ring of flavonoids or catechins improved their tyrosinase inhibition, which may also be correlated with their enhanced antioxidant activity [48]. Thus, the occurrence of phenolics with multiple hydroxyl groups, such as myricetin-3-*O*-rhamnoside and kaempferol-*O*-hexoside, can be related to the high tyrosinase inhibition observed. Overall, our results suggested that the extract of *L. spathulatum* contains molecules with tyrosinase inhibitory properties, most probably phenolic acids and flavonoids, and therefore, with interest for the cosmetic, pharmaceutical, agriculture, and food sectors. To our knowledge, this is the first time a tyrosinase inhibition capacity of *L. spathulatum* has been reported.

Diabetes mellitus (DM) is a chronic disorder with two main types: type-1 (T1DM) and type-2 (T2DM), the latter accounting for approximately 90% of cases. Postprandial hyperglycemia may increase the risk of T2DM and subsequent complications, including cardiovascular disease and nephropathy [49]. Glycemic control is therefore vital to manage hyperglycemia and may be accomplished by inhibiting carbohydrate hydrolyzing enzymes, such as α-glucosidase, by the use of enzyme inhibitors, for example, acarbose [50]. However, such compounds exhibited some undesirable side effects, including abdominal and liver disorders [51], thus urging the need to identify new α-glucosidase inhibitors with reduced secondary effects. In this study, the extracts had a significant capacity to inhibit α-glucosidase, especially the hydroethanolic extract (Table 1). The inhibitory capacity of the extracts was significantly higher than that of acarbose. The same results were reported for other *Limonium* species, such as *L. boitardii* [52] and *L. algarvense* [29], and other halophytes, such as *Polygonum maritimum* [53].

Some polyphenolic compound groups, such as phenolic acids and flavonoids, exhibit strong inhibition towards α-glucosidase, and it was found that the position and number of hydroxyl groups in the flavonoids (mainly C5-OH, C6-OH, and C7-OH groups in A-ring and the three B-ring OH groups) are determining factors for α-glucosidase inhibition [54]. Myricetin and epigallocatechin gallate were reported as strong inhibitors, while quercetin and epigallocatechin have weaker activity [54]. Some molecules identified only in the hydroethanolic extract, such as riboflavin, myricetin derivatives, and prodelphinidin, display inhibition properties in carbohydrate hydrolyzing enzymes and, therefore, hold potential as antidiabetic drugs [54,55].

Obesity is a worldwide problem and an important factor risk for the onset of several health problems, including cardiovascular diseases, hypertension, and diabetes, to name a few [56]. Pancreatic lipase directly affects the absorption of fatty acids (FA) in the intestine; it is the primary lipase secreted from the pancreas and hydrolyzes dietary lipids in the digestive system, converting triacylglycerol substrates in ingested oils to monoglycerides and free FA [56]. Lipase inhibitors inhibit the absorption of FA and reduce its accumulation in the body, therefore reducing the level of LDL (low-density lipoprotein) in the serum and increasing the level of HDL (high-density lipoprotein) [56]. Lipase inhibitors are thus an important therapeutical strategy to control obesity. In this study, the extracts did not inhibit lipase (Table 1). The same result was observed in methanol extracts of *L. effusum*, while relevant lipase inhibition was reported in *L. quesadense* [46], *L. sinuatum* [57], *L. contortirameum,* and *L. virgatum* [58].

### 2.2. Cytotoxic Properties

While it is widely acknowledged that natural products are generally deemed safe, emerging evidence indicates potential safety issues associated with these substances [59].

It is, therefore, of utmost importance to evaluate the possible toxic effects of herbal products by, for example, assessing their acute toxicity toward mammalian cell lines [59]. In this study, we determined the cytotoxic properties of *L. spathulatum* extracts towards a panel of cell lines, including tumor cells (stomach gastric adenocarcinoma: AGS; colorectal adenocarcinoma: CaCo2; human breast carcinoma: MCF-7 and non-small cell lung cancer: NCI-H460), as well as on cell lines from non-tumor origin (porcine liver primary culture: PLP2 and monkey kidney epithelial cells: VERO). Our results indicated that *L. spathulatum* extracts have low to moderated cytotoxic activity since significantly higher GI_50_ values were observed compared with the positive controls (Table 2).

The water extract exhibited the highest cytotoxic activity towards the VERO cell line, with a GI_50_ value of 23 µg/mL (Table 2). Although raising some questions regarding its safe use, this result could be further explored, in view of the criteria (GI_50_ below 30 μg/mL) established by the American National Cancer Institute NCI for the selection of crude extracts to be pursued in the search for potential anti-tumoral leads [60]. Nevertheless, it is important to consider that the techniques employed here serve as an initial assessment of toxicity and are meant to guide subsequent investigations conducted on mammalian animal models, which are currently underway. As far as we know, this is the first report of the in vitro toxicity assessment of *L. spathulatum* extracts.

### 2.3. Chemical Composition of the Extracts

The extracts were analyzed by HPLC-ESI-MS/MS, and the results are summarized in Table 3. The chemical profile of the ethanol extract was previously reported [23]. The hydroethanolic extract had the highest number of identified compounds (64), followed by the ethanol (58) and the water extracts (45). Similar molecules were identified in all samples and were mainly hydroxybenzoic acids (gallic and syringic acid), hydroxycinnamic acids (caffeic, coumaric, and ferulic acids), and flavonoids (catechin and epigallocatechin). However, some compounds were specific to a particular extract. For example, prodelphinidin B (polymeric tannin) and uralenneoside (p-hydroxybenzoic acid alkyl ester), or isomer sulfate, were only detected in the water extracts. These molecules were previously identified in hydroalcoholic extracts from *L. gmelinii* and were suggested as the main contributors to the antiradical potential and antiexudative properties of the samples [61]. Prodelphinidin A gallate (proanthocyanidin), ethyl gallate (galloyl ester, ethyl ester of gallic acid), riboflavin (vitamin B2), and the flavonoids myricetin-*O*-galloylhexoside, myricetin-*O*-(di-*O*-acetyl) rhamnoside isomer 1, myricetin-*O*-(di-*O*-acetyl) rhamnoside isomer 2 and dihydroxy-trimethoxy(iso)flavone, were only detected in the hydroethanolic extract. Among these compounds, uralenneoside was reported here for the first time in this genus. Uralenneoside is a phenolic acid derivative identified in extracts from different plant species, such as *Pistacia terebinthus* L. [62] and *Desmodium tortuosum*, and exhibits antioxidant and enzyme inhibitory properties [63]. Other compounds were described in *L. boitardii* [52], namely gallic acid, epigallocatechin-3-*O*-gallate (Teatannin II), rutin (quercetin-3-*O*-rutinoside), myricetin (3,3′,4′,5,5′,7-Hexahydroxyflavone), and quercetin (3,3′,4′,5,7-Pentahydroxyflavone). Other metabolites were previously detected in other *Limonium* species, including quinic acid in aerial parts of *L. tubiflorum* var *tubiflorum* with antioxidant, antimicrobial, and anti-HIV-1 assets [64]. Shikimic acid was detected in all extracts from *L. spathulatum* and is the precursor for the synthesis of oseltamivir (Tamiflu), the only drug against avian flu caused by the H5N1 virus [65]. Myricetin and its derivatives (myricetin-*O*-galloylhexoside, myricetin-*O*-(di-*O*-acetyl)rhamnoside isomer 1, and myricetin-*O*-(di-O-acetyl) rhamnoside isomer 2) exhibited antioxidant, anticarcinogenic, antiviral and antimicrobial properties and were identified in aerial parts of *L. sinuatum* [66], while myricetin-3-*O*-rutinoside was previously identified in *L. algarvense*’s water extracts [67]. Prodelphinidin A gallate and ethyl gallate were previously detected in *L. gmelinii* roots [61], while chlorogenic acid, gallic acid, and rutin were identified in the shoot extracts of *L. delicatulum* [46]. Moreover, epigallocatechin gallate, phlorizin, phloretin, and quercetin were also detected in aqueous extracts of *L. contortirameum* and *L. virgatum* [58], while tannic acid and hyperoside were quantified in the ethyl acetate fractions of aerial organs *L. effusum* and *L. sinuatum* [57].

## 3. Materials and Methods

### 3.1. Chemicals

All the chemicals used in this study were of analytical grade. Fischer Scientific (Loughborough, UK) supplied acetylcholinesterase from electrical eel (AChE, EC 3.1.1.7) butyrylcholinesterase from equine serum (BChE, EC 3.1.1.8), butyrylthiocholine iodide, galanthamine, acetylthiocholine iodide, 5,5-dithiobis(2-nitrobenzoic) acid (DTNB), lipase from porcine pancreas, (EC 3.1.1.3), tyrosinase from mushroom (EC 1.14.18.1), and α-glucosidase from *Saccharomyces cerevisiae* (EC 232-604-7). Ethanol was purchased from Riedel de Haën (Buchs, Switzerland). Merck (Darmstadt, Germany) delivered ethanol, methanol, and formic acid. Other solvents and chemicals were provided by VWR International (Leuven, Belgium).

### 3.2. Plant Material

Leaves of *L. spathulatum* were collected from adult plants in coastal areas of Tabarka in Tunis (Tunisia) (coordinates: 36°57′23″ N8°45′28.5″ E) in March 2019 during the flowering season. Leaves were taken to the laboratory, cleaned with tap water, dehydrated at 37 °C for one week, reduced to powder, and stored in the dark at 4 °C until use in the assays. The taxonomical identification was made by Dr. Abderrazek Smaoui (Center of Biotechnology of Borj Cedria, CBBC, Tunisia), and a voucher specimen (voucher code LPEH01) can be found in the herbarium of the Laboratory of Extremophile Plants of CBBC.

### 3.3. Preparation of the Extracts

The extracts were prepared by mixing dried leaves with ethanol (100% and 50%, *w*/*w*) and water (1:40, *w*/*w*) and extracted overnight at room temperature (RT) with stirring. The extracts were filtered (Whatman paper no. 4), and the water extracts were freeze-dried, while the solvents of the organic extracts were removed in a rotary evaporator at reduced temperature and pressure. The dried residues were weighed, dissolved in the corresponding solvent (50 mg/mL), and stored (−20 °C) until analysis.

### 3.4. Enzymatic Inhibition

In all the assays, the extracts were tested at concentrations ranging from 0.009 to 5 mg/mL. Results were calculated considering the negative control used, containing the corresponding solvent, and were expressed as inhibitory median concentrations (IC_50_ values, mg/mL).

#### 3.4.1. Inhibition of AChE and BChE

The AChE and BChE inhibitory activities were determined using Ellman’s method [68]. Galantamine was used as the positive control at concentrations up to 1 mg/mL.

#### 3.4.2. Inhibition of Baker’s Yeast α-Glucosidase

The inhibitory activity of the extracts was appraised against α-glucosidase from *Saccharomyces cerevisiae* by a previously described method [69]. Acarbose was used as the positive control at concentrations up to 10 mg/mL.

#### 3.4.3. Inhibition of Tyrosinase

The inhibitory activity against tyrosinase was determined according to a protocol described by [70]. Arbutin was used as the positive control at concentrations up to 1 mg/mL.

#### 3.4.4. Inhibition of Lipase from Porcine Pancreas

The inhibitory activity against lipase from the porcine pancreas was evaluated according to the method described by [71]. Orlistat was used as the positive control at concentrations up to 1 mg/mL.

### 3.5. In Vitro Toxicological Evaluation

Tumor cell lines (human gastric: AGS; colorectal: Caco2; breast: MCF-7; and lung: NCI-H460) and non-tumor cell lines (monkey kidney: Vero; and primary pig liver culture: PLP2) were used to evaluate the in vitro acute toxicity of the extracts. AGS, Caco2, Vero, and RAW264.7 cells were purchased from the European Collection of Authenticated Cell Cultures (ECACC), while MCF-7 and NCI-H460 cells were provided by Leibniz-Institute DSMZ. All cell lines were seeded at a density of 10,000 cells/well, except for Vero cells, which were prepared at 19,000 cells/well. According to the procedure previously described by [70], 190 µL of cell suspension was added to 10 µL of different concentrations of extract solutions and kept for 60 min at room temperature and incubated at 37 °C for 72 h. Then, Sulforhodamine B (SRB) colorimetric assay for the cytotoxicity screening of compounds to adherent cells was applied. For this, 100 µL of cold 10% (*w*/*v*) TCA was added to the wells, and the plates were incubated at 4 °C for 1 h. Next, the TCA was removed, and the adhered cells were washed three times with water and dried. Cells were then stained with 100 µL of 0.057% (*w*/*v*) SRB solution for 30 min at room temperature. Excess dye was removed by washing three times with 1% (*v*/*v*) acetic acid. Once the plates were dried, 200 µL of 10 mM Tris base was used to dissolve the cells, and the absorbance at 540 nm of protein-bound dye was measured in a microplate reader Biotek ELX800 (Biotek Instrument Inc., Winooski, VT, USA). For each cell line, plated cells without extracts were used as a negative control. In addition, the antitumor drug ellipticine was used as a positive control. Results were expressed as GI_50_ values (extract concentration capable of inhibiting 50% of cell growth).

### 3.6. High-Performance Liquid Chromatography Coupled with Electrospray Ionization Mass Spectrometry (HPLC-ESI-MS/MS) Analysis

The chemical composition of the extracts was determined using a Dionex Ultimate 3000RS UHPLC instrument (Thermo Fischer Scientific Inc., New York, NY, USA). Samples were filtered (0.22 μm PTFE filter membrane, Labex Ltd., Budapest, Hungary) before HPLC analysis and injected onto a Thermo Accucore C18 (100 mm × 2.1, mm i. d., 2.6 μm) column thermostated at 25 °C (±1 °C). The solvents used were water (A) and methanol (B), acidified with 0.1% formic acid, and the flow rate was maintained at 0.2 mL/min. A gradient elution was used: 5% B (0–3 min), a linear gradient increasing from 5% B to 100% (3–43 min), 100% B (43–61 min), a linear gradient decreasing from 100% B to 5% (61–62 min) and 5% B (62–70 min). The column was coupled with a Thermo Q-Exactive Orbitrap mass spectrometer (Thermo Scientific, Waltham, MA, USA) equipped with an electrospray ionization source. Spectra were recorded in positive and negative-ion mode, respectively. The trace finder 3.1 (Thermo Scientific, USA) software was applied for target screening. Most of the compounds were identified based on previously published studies or data found in the literature. The exact molecular mass, isotopic pattern, characteristic fragment ions, and retention time were always used to identify the molecules.

### 3.7. Statistical Analysis

Experiments were conducted at least in triplicate, and results were expressed as mean ± standard deviation (SD). Differences in significance (*p* < 0.05) were evaluated by one-way analysis of variance (ANOVA), pursued by the Tukey HSD test. Statistical analyses were performed using XLStat2014^®^. The IC_50_ and GI_50_ values were determined by sigmoidal fitting of these data in the GraphPad Prism v. 5.0 software.

## 4. Conclusions

Our results indicate for the first time that *L. spathulatum* collected from Tunisian sea cliffs contains compounds that inhibit BChE, α-glucosidase, and tyrosinase and display nil or moderate toxicity towards mammal cell lines. The HPLC-ESI-MS/MS analysis of the extracts allowed for the identification of a high number of bioactive metabolites, mainly phenolics and flavonoids, which may be related to the detected bioactivities. These results strongly suggest that *L. spathulatum* should be further explored as a source for high-value-added products.

## Figures and Tables

**Table 1 plants-12-03391-t001:** Enzymatic inhibitory activity on acetylcholinesterase (AChE), butyrylcholinesterase (BChE), α-glucosidase tyrosinase, and lipase of hydroethanolic, ethanol, and water extracts of *Limonium spathulatum.* Results are expressed as IC_50_ values (mg/mL).

Extract	AChE	BChE	α-Glucosidase	Tyrosinase	Lipase
Ethanol	1.75 ± 0.06 ^c^	0.27 ± 0.09 ^c^	0.16 ± 0.01 ^a^	0.34 ± 0.01 ^a^	na
Water	0.23 ± 0.04 ^b^	0.06 ± 0.02 ^b^	0.16 ± 0.03 ^a^	1.10 ± 0.04 ^c^	na
Hydroethanolic	0.31 ± 0.05 ^b^	0.03 ± 0.01 ^a^	0.04 ± 0.01 ^a^	1.91 ± 0.55 ^c^	na
Positive controls					
Galantamine	0.01 ± 0.00 ^a^	0.31 ± 0.03 ^c^	-	-	-
Acarbose	-	-	3.14 ± 0.23 ^b^	-	-
Orlistat	-	-	-	-	0.11 ± 0.02
Arbutin	-	-	-	0.17 ± 0.01 ^b^	-

Values represent the mean ± standard deviation (SD) of at least three experiments performed in triplicate (n = 9). Different letters (a–c) in the same column indicate significant differences between extracts by Tukey’s HSD test (*p* < 0.05). -: not tested; na: no activity detected.

**Table 2 plants-12-03391-t002:** Cytotoxic activity of hydroethanolic, ethanol, and water extracts of *Limonium spathulatum*. Results are expressed as GI_50_ values (µg/mL), which correspond to the extract concentration responsible for 50% of cell growth inhibition.

	Hydroethanolic	Ethanol	Water	Positive Control
Cytotoxicity				
Tumor cells				Ellipticine
AGS (gastric adenocarcinoma)	93 ± 5	63 ± 2	42 ± 4	1.23 ± 0.03
Caco2 (colorectal adenocarcinoma)	72 ± 5	40 ± 4	90 ± 1	1.21 ± 0.02
MCF-7 (human breast carcinoma)	60 ± 1	177 ± 12	49 ± 3	1.21 ± 0.02
NCI-H460 (non-small cell lung cancer)	42 ± 1	83 ± 4	40 ± 1	0.9 ± 0.1
Non-tumor cells				Ellipticine
PLP2 (porcine liver primary culture)	>400	79 ± 8	40 ± 2	1.4 ± 0.1
VERO (monkey kidney epithelial cells)	55 ± 3	64 ± 2	23 ± 2	1.41 ± 0.06

**Table 3 plants-12-03391-t003:** High-performance liquid chromatography coupled with electrospray ionization mass spectrometry (HPLC-ESI-MS/MS) tentative identification of metabolites present in hydroethanolic, ethanol, and water extracts of *Limonium spathulatum*. * Results published in Youssef et al. (2022) [23].

	Formula	RT	[M + H]^+^	[M − H]^–^	Hydroethanolic Extract	Water Extract	Ethanol Extract *
Quinic acid	C7H12O6	2.11		191.05557	+	-	+
Shikimic acid	C7H10O5	2.16		173.04500	+	+	+
Galloylhexose	C13H16O10	2.87		331.06653	+	+	+
Gallic acid (3,4,5-Trihydroxybenzoic acid)	C7H6O5	3.18		169.01370	+	+	+
Prodelphinidin B	C30H26O14	4.39		609.12444	-	+	-
Gallocatechin (Gallocatechol)	C15H14O7	5.63		305.06613	+	+	+
Coumaroylhexose sulfate isomer 1	C15H18O11S	7.79		405.04916	+	+	+
Caffeoylhexose sulfate isomer 1	C15H18O12S	9.00		421.04408	+	+	+
Uralenneoside or isomer	C12H14O8	11.03		285.06105	+	+	+
Caffeoylhexose	C15H18O9	11.81		341.08726	+	+	+
Coumaroylhexose sulfate isomer 2	C15H18O11S	12.22		405.04916	+	+	+
Caffeoylhexose sulfate isomer 2	C15H18O12S	12.80		421.04408	+	+	+
Epigallocatechin (Epigallocatechol)	C15H14O7	13.45		305.06613	+	+	+
Prodelphinidin A gallate	C37H28O18	13.59		607.10879	+	-	-
Chlorogenicacid (3-*O*-Caffeoylquinic acid)	C16H18O9	14.42	355.10291		+	+	+
Coumaroylhexose isomer 1	C15H18O8	14.46		325.09235	+	+	+
Uralenneoside or isomer sulfate	C12H14O11S	14.53		365.01786	-	+	-
Caffeic acid	C9H8O4	14.60		179.03444	+	+	+
Biflorin	C16H18O9	14.78	355.10291		+	+	+
Digalloylhexose	C20H20O14	14.98		483.07749	+	+	+
Coumaroylhexose isomer 2	C15H18O8	15.16		325.09235	+	+	+
Isobiflorin	C16H18O9	15.56	355.10291		+	+	+
Epigallocatechin-3-*O*-gallate (Teatannin II)	C22H18O11	16.25		457.07709	+	+	+
Dihydrokaempferol-*O*-hexoside	C21H22O11	17.18		449.10839	+	+	+
Ethyl gallate	C9H10O5	17.65		197.04500	+	-	-
Coumaroyl-hexosylglycerate	C18H22O11	18.09		413.10839	+	+	+
Riboflavin	C17H20N4O6	18.46	377.14611		+	-	-
Isololiolide	C11H16O3	18.63	197.11777		+	+	+
Myricetin-*O*-galloylhexoside	C28H24O17	19.04		631.09353	+	-	-
Ferulic acid	C10H10O4	19.36		193.05009	+	+	+
Unidentified alkaloid	C13H12N2O3	19.55	245.09262		+	+	+
Loliolide	C11H16O3	19.84	197.11777		+	+	+
Myricetin-*O*-hexoside	C21H20O13	20.37		479.08257	+	+	+
Myricetin-3-*O*-rutinoside	C27H30O17	21.05		625.14048	+	+	+
Myricetin-*O*-pentoside	C20H18O12	21.50		449.07201	+	+	+
Myricitrin (Myricetin-3-*O*-rhamnoside)	C21H20O12	21.68		463.08765	+	+	+
N-cis-Feruloyltyramine	C18H19NO4	22.35	314.13924		+	+	+
Hyperoside or Isoquercitrin	C21H20O12	22.31		463.08765	+	+	+
Rutin (quercetin-3-*O*-rutinoside)	C27H30O16	22.60		609.14557	+	+	+
Coatline A or isomer	C21H24O10	22.74		435.12913	+	+	+
Methoxy-pentahydroxy(iso)flavone-*O*-hexoside	C22H22O13	22.87		493.09822	+	+	+
Myricetin (3,3′,4′,5,5′,7-Hexahydroxyflavone)	C15H10O8	23.80		317.02974	+	+	+
Kaempferol-7-*O*-glucoside	C21H20O11	23.84		447.09274	+	+	+
Phlorizin	C21H24O10	24.05		435.12913	+	+	+
Quercitrin (Quercetin-3-*O*-rhamnoside)	C21H20O11	24.21		447.09274	+	+	+
Astragalin (Kaempferol-3-*O*-glucoside)	C21H20O11	24.41		447.09274	+	+	+
Kaempferol-3-O-rutinoside (Nicotiflorin)	C27H30O15	24.54		593.15065	+	+	+
N-trans-Feruloyltyramine	C18H19NO4	24.60	314.13924		+	+	+
Dimethoxy-tetrahydroxy(iso)flavone isomer 1	C17H14O8	25.79		345.06104	+	+	+
Afzelin (Kaempferol-3-*O*-rhamnoside)	C21H20O10	26.19		431.09782	+	+	+
Myricetin-*O*-(di-*O*-acetyl)rhamnosideisómer 1	C25H24O14	26.54		547.10879	+	-	-
Dihydroactinidiolide	C11H16O2	26.66	345.09743		+	+	+
Quercetin (3,3′,4′,5,7-Pentahydroxyflavone)	C15H10O7	26.71		301.03483	+	+	+
Naringenin (4′,5,7-Trihydroxyflavanone)	C15H12O5	27.23		271.06065	+	+	+
Myricetin-*O*-(di-*O*-acetyl)rhamnosidesómer 2	C25H24O14	27.65		547.10879	+	-	-
Quercetin-3-*O*-methyl ether	C16H12O7	28.10		315.05048	+	+	+
Phloretin (dihydronaringenin)	C15H14O5	28.23		273.07630	+	+	+
Dimethoxy-tetrahydroxy(iso)flavone isomer 2	C17H14O8	28.34		345.06104	+	+	+
Trihydroxy-trimethoxy(iso)flavone isomer 1	C18H16O8	30.37		359.07670	+	+	+
Trihydroxy-trimethoxy(iso)flavone isomer 2	C18H16O8	31.10		359.07670	+	+	+
Malyngic acid or isomer	C18H32O5	32.30		327.21715	+	+	+
Trihydroxy-trimethoxy(iso)flavone isomer 3	C18H16O8	32.63		359.07670	+	+	+
Dimethoxy-trihydroxy(iso)flavones	C17H14O7	32.85		329.06613	+	+	+
Dihydroxy-tetramethoxy(iso)flavones	C19H18O8	33.26		373.09235	+	+	+
Pinellic acid	C18H34O5	33.61		329.23280	+	+	+
Dihydroxy-trimethoxy(iso)flavones	C18H16O7	35.03	345.09743		+	-	-

+: Detected; -: Not detected; RT: retention time; [M + H]^+^: positive ion mode; [M − H]^−^: negative ion mode.

## Data Availability

The dataset is available upon request from the corresponding author.

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
