# Peer review of "Harnessing the Bioactive Potential of Limonium spathulatum (Desf.) Kuntze: Insights into Enzyme Inhibition and Phytochemical Profile"

_plants, 2023, doi:10.3390/plants12193391_

Round 1

Reviewer 1 Report

The manuscript entitled:  Harnessing the Bioactive Potential of Limonium spathulatum (Desf.) 2 Kuntze: Insights into Enzyme Inhibition and phytochemical profile is an experimental work focused on the evaluation of the Limonium spathulatum  extracts on selected selected enzymes. The authors reliably reviewed the available new scientific literature. In this regard, I appreciate the author’s work. Unfortunately, there are several serious errors and shortcomings in the manuscript.

1)     1) The language of the work is often incomprehensible, there are a lot of grammatical, and linguistic errors in the work, which strongly make it difficult to read the manuscript;

2)     What is the difference between food grade extracts and “just” extracts? That should be explained;

3)     On what criteria did the authors select the enzymes for the study presented in this paper?

4)     What criteria were used to select the cell lines on which the extracts were tested? If we are testing activity against Acetylcholinesterase it would be reasonable to test the safety of the extracts against central nervous system cells. If we are testing activity against tyrosinase - it would be reasonable to test the safety of the extracts against skin cell lines.

5)     In Table 2 the unit of the cytotoxic effect should be given;

6)     The data in Table 3 replicates the results of previous studies (with minor changes);

7)     The origin of the tyrosinase and lipase should be given;

The native speaker must linguistically correct the manuscript.

Author Response

The manuscript entitled:  Harnessing the Bioactive Potential of Limonium spathulatum (Desf.) 2 Kuntze: Insights into Enzyme Inhibition and phytochemical profile is an experimental work focused on the evaluation of the Limonium spathulatum extracts on selected selected enzymes. The authors reliably reviewed the available new scientific literature. In this regard, I appreciate the author’s work. Unfortunately, there are several serious errors and shortcomings in the manuscript.

Response: Thank you for reviewing the Mn. Please find below our responses to your comments

1) The language of the work is often incomprehensible, there are a lot of grammatical, and linguistic errors in the work, which strongly make it difficult to read the manuscript;

Response: The authors acknowledge this comment. The paper was deeply revised in terms of grammar and spelling.

2) What is the difference between food grade extracts and “just” extracts? That should be explained;

Response: Extracts are those prepared by any solvent, while food grade are those by using food grade solvents, such as ethanol and water (Directive 2009/32/EC of the European Parliament and of the Council of 23 April). This information was introduced in the text (Introduction).

3) On what criteria did the authors select the enzymes for the study presented in this paper?

Response: Enzymes were selected based on their relevance for the target diseases, as explained on the “Introduction” and “Results and Discussion” sections.

4) What criteria were used to select the cell lines on which the extracts were tested? If we are testing activity against Acetylcholinesterase it would be reasonable to test the safety of the extracts against central nervous system cells. If we are testing activity against tyrosinase - it would be reasonable to test the safety of the extracts against skin cell lines.

Response: The selected cell lines are part of a panel of mammalian cell lines available in our lab to perform cytotoxicity studies. Despite not being specific to the target assays, they address the sensitivity of mammalian cell lines to possible toxic effects of the extracts, delivering reliable and quick results and reducing in vivo testing (Rodrigues et al., 2016; Saad et al., 2006).

5) In Table 2 the unit of the cytotoxic effect should be given;

Response: This information was introduced in the table caption.

6) The data in Table 3 replicates the results of previous studies (with minor changes);

Response: Data on table 3 regarding results for the ethanol extract were published previously in the paper by Youssef et al., 2022. This is clearly indicated in the table caption.

7) The origin of the tyrosinase and lipase should be given;

Response:  This information was introduced in 4. Materials and Methods, 4.1. Chemicals

Reviewer 2 Report

plants-2596800

Abstract

Line 22  authors must decide the use  of neurodegeneration  or  Alzheimer´s disease (AD) in agreement  with introductions (lines 51 and 92)

Line 22 (acetylcholinesterase- AChE, and butyrylcholinesterase -BChE) change as [acetylcholinesterase (AChE) and butyrylcholinesterase (BuChE)]

Line 24 hyperpigmentation  dont include enzime please include

Keywords: 

-include the name of the plant used  Limonium spathulatum delete halophytes

- delete cholinesterase inhibitors or include all enzimes inhibitors such  glucosidase inhibitors, tyrosinase inhibiitors, lipase

Lines 27-29  said "were more effective "  dont said against which enzime   

Introduction

Lines 50-51 change (glucosidase , amylase) as (a-glucosidase and amylase)

Line 51 change (acetyl- and butyrilcholinesterase (AChE and BChE) by [acetyl- and butyrilcholinesterase (AChE and BuChE)]

Line 68 change "(e.g., superoxide dismutase, glutathione peroxidase; [11]) as "(......dismutase and glutathione peroxidase ) [11]

Line 69 change  (e.g., phenolics, flavonoids; [12,13,14,15]) as (e.g., phenolics and flavonoids) [12-15] or (e.g., polyphenols) [12-15].

Line 92  change (AChE, BuChE) as (AChE and BuChE); (a-glucosidase  and amylase)

In hyperpigmentation please icnlude which enzime

Results

In Table 1 What meaning Na  please include 

line 104 decide in all text of manuscript  the use "butyryl-cholinesterase" or butyrylcholinesterase

Line 244 and check in all manuscript L. sphathulatum change to italics as L. sphathulatum

Lines 257 and 259 are correct GI50?. Also The Table 3 dont contain GI50 values

plants-2596800

Abstract

Line 22  authors must decide the use  of neurodegeneration  or  Alzheimer´s disease (AD) in agreement  with introductions (lines 51 and 92)

Line 22 (acetylcholinesterase- AChE, and butyrylcholinesterase -BChE) change as [acetylcholinesterase (AChE) and butyrylcholinesterase (BuChE)]

Line 24 hyperpigmentation  dont include enzime please include

Keywords: 

-include the name of the plant used  Limonium spathulatum delete halophytes

- delete cholinesterase inhibitors or include all enzimes inhibitors such  glucosidase inhibitors, tyrosinase inhibiitors, lipase

Lines 27-29  said "were more effective "  dont said against which enzime   

Introduction

Lines 50-51 change (glucosidase , amylase) as (a-glucosidase and amylase)

Line 51 change (acetyl- and butyrilcholinesterase (AChE and BChE) by [acetyl- and butyrilcholinesterase (AChE and BuChE)]

Line 68 change "(e.g., superoxide dismutase, glutathione peroxidase; [11]) as "(......dismutase and glutathione peroxidase ) [11]

Line 69 change  (e.g., phenolics, flavonoids; [12,13,14,15]) as (e.g., phenolics and flavonoids) [12-15] or (e.g., polyphenols) [12-15].

Line 92  change (AChE, BuChE) as (AChE and BuChE); (a-glucosidase  and amylase)

In hyperpigmentation please icnlude which enzime

Results

In Table 1 What meaning Na  please include 

line 104 decide in all text of manuscript  the use "butyryl-cholinesterase" or butyrylcholinesterase

Line 244 and check in all manuscript L. sphathulatum change to italics as L. sphathulatum

Lines 257 and 259 are correct GI50?. Also The Table 3 dont contain GI50 values

Check all manuscript

Author Response

Abstract

1) Line 22  authors must decide the use  of neurodegeneration  or  Alzheimer´s disease (AD) in agreement  with introductions (lines 51 and 92)

Response: The text was corrected.

2) Line 22 (acetylcholinesterase- AChE, and butyrylcholinesterase -BChE) change as [acetylcholinesterase (AChE) and butyrylcholinesterase (BuChE)]

Response: The text was corrected.

3) Line 24 hyperpigmentation  dont include enzime please include

Response: Tyrosinase is implicated in both hyperpigmentation and food oxidation

Keywords

1) include the name of the plant used  Limonium spathulatum delete halophytes 

Response: We don’t agree with this. Keywords should include words not included in the title of the manuscript, to facilitate finding the work when searching on the web. So, we included instead the vernacular name of Limonium: sea lavender.

2) delete cholinesterase inhibitors or include all enzimes inhibitors such  glucosidase inhibitors, tyrosinase inhibiitors, lipase

Response: Keywords were modified according to this comment.

3) Lines 27-29  said "were more effective "  dont said against which enzyme

Response: The text was corrected.   

Introduction

1) Lines 50-51 change (glucosidase , amylase) as (a-glucosidase and amylase)

2) Line 51 change (acetyl- and butyrilcholinesterase (AChE and BChE) by [acetyl- and butyrilcholinesterase (AChE and BuChE)]

3) Line 68 change "(e.g., superoxide dismutase, glutathione peroxidase; [11]) as "(......dismutase and glutathione peroxidase ) [11]

4) Line 69 change  (e.g., phenolics, flavonoids; [12,13,14,15]) as (e.g., phenolics and flavonoids) [12-15] or (e.g., polyphenols) [12-15].

Response: The text was corrected.   

5) Line 92  change (AChE, BuChE) as (AChE and BuChE); (a-glucosidase  and amylase)

In hyperpigmentation please icnlude which enzyme

Response: The text was corrected. Tyrosinase is implicated in both hyperpigmentation and food oxidation.

Results

1) In Table 1 What meaning Na  please include 

2) line 104 decide in all text of manuscript  the use "butyryl-cholinesterase" or 3) butyrylcholinesterase

3) Line 244 and check in all manuscript L. sphathulatum change to italics as L. sphathulatum

4) Lines 257 and 259 are correct GI50?. Also The Table 3 dont contain GI50 values

Response: All the corrections were made.   

Round 2

Reviewer 1 Report

Unfortunately, there are still several shortcomings in the manuscript.

Abstract:

Line 34 – should be: tyrosinase (not tirosinase)

Line 35 – should be: Tested samples did not inhibit lipase..(not: Samples were not able to inhibition activity towards lipase)

Lines 171-172 - Please change the order of cosmetic ingredients - hydroquinone is the only one that can be obtained synthetically

Line 275 – The authors rather refer to the data in Table 2.

Lines 327 – 328 – I cannot find the information the authors gave me in the response to previous comments (“Data on table 3 regarding results for the ethanol extract were published previously in the paper by Youssef et al., 2022. This is clearly indicated in the table caption.”) There is no information that those data were already published.

Starting with item 24 in the bibliography - authors should bold the year of publication of each reference article

Bibliographic position 32 - please change font size and size

The language style still needs to be checked

Author Response

Responses to reveiwer's comments

We want to thank the Reviewer for their comments that helped us to improve the manuscript. We have tried our best to improve the manuscript. Below you can find our responses to reviewer comments.

1. Comments and Suggestions for Authors

Unfortunately, there are still several shortcomings in the manuscript.

Response: We revised again the manuscript according to your comments and suggestions

Abstract:

Line 34 – should be: tyrosinase (not tirosinase)

Line 35 – should be: Tested samples did not inhibit lipase..(not: Samples were not able to inhibition activity towards lipase)

Response: Done

Lines 171-172 - Please change the order of cosmetic ingredients - hydroquinone is the only one that can be obtained synthetically

Response: Done

Line 275 – The authors rather refer to the data in Table 2.

Response: Yes, we corrected this mistake.

Lines 327 – 328 – I cannot find the information the authors gave me in the response to previous comments (“Data on table 3 regarding results for the ethanol extract were published previously in the paper by Youssef et al., 2022. This is clearly indicated in the table caption.”) There is no information that those data were already published.

Response: You are right. You can find now this information in the table caption.  

Starting with item 24 in the bibliography - authors should bold the year of publication of each reference article

Bibliographic position 32 - please change font size and size

Response: We agree. Reference citations are carefully checked.

Comments on the Quality of English Language

The language style still needs to be checked

Response: The language has been checked again by an English fluent speaker.